# More Causes Less Effect: Destructive Interference in Decision Making

**DOI:** 10.3390/e24050725

**Published:** 2022-05-20

**Authors:** Irina Basieva, Vijitashwa Pandey, Polina Khrennikova

**Affiliations:** 1International Center for Mathematical Modeling in Physics and Cognitive Science, Linnaeus University, S-35195 Växjö, Sweden; 2Industrial and Systems Engineering Department, Oakland University, Rochester, MI 48309, USA; pandey2@oakland.edu; 3School of Business, University of Leicester, Leicester LE1 7RH, UK; pk228@leicester.ac.uk

**Keywords:** customer decision making, product design, non-classical information processing, interference

## Abstract

We present a new experiment demonstrating destructive interference in customers’ estimates of conditional probabilities of product failure. We take the perspective of a manufacturer of consumer products and consider two situations of cause and effect. Whereas, individually, the effect of the causes is similar, it is observed that when combined, the two causes produce the opposite effect. Such negative interference of two or more product features may be exploited for better modeling of the cognitive processes taking place in customers’ minds. Doing so can enhance the likelihood that a manufacturer will be able to design a better product, or a feature within it. Quantum probability has been used to explain some commonly observed “non-classical” effects, such as the disjunction effect, question order effect, violation of the sure-thing principle, and the Machina and Ellsberg paradoxes. In this work, we present results from a survey on the impact of multiple observed symptoms on the drivability of a vehicle. The symptoms are assumed to be conditionally independent. We demonstrate that the response statistics cannot be directly explained using classical probability, but quantum formulation easily models it, as it allows for both positive and negative “interference” between events. Since quantum formalism also accounts for classical probability’s predictions, it serves as a richer paradigm for modeling decision making behavior in engineering design and behavioral economics.

## 1. Introduction

We present the results of an experimental study demonstrating that the conjunction of two causes, a and b, each individually in favor of event d, can together decrease the probability of d. A study by Zheng et al., [1] shows that prior experience interacts with causal information in a non-trivial way. In our study, we show that multiple causes of the same type can also interplay in a way not supported by the classical information-processing paradigm. It is well-known that causal information can aid decision making by providing supporting reasons for a choice, clarifying the valuation of options [2,3,4]. However, as we show in this paper, causation in decision making can have peculiarities such as destructive or constructive interference of the causes. To mathematically model this interference of probabilities, we construct a quantum probabilistic model. The use of quantum-like modeling is also supported by theoretical classical probabilistic analysis.

Motivation for our experimental study came from the area of design of complex systems or products, such as an automobile or an airplane. Such design-projects, which are inherently decision-making processes, involve balancing many characteristics ranging from technical features to manufacturability, while keeping the voice of the customer in mind [5,6,7]. Design engineers themselves have preferences which reflect themselves in the design-decisions they make, both in terms of tradeoffs between attributes and the various decisions under uncertainty. The more aligned these preferences are with the voice of the customer, the higher the likelihood of the success of the product is. Thus, the psychology of designers and customers has to be seriously taken into account in product development projects.

Notwithstanding survey design issues, it may be tempting to ascribe, or even require, mathematical consistency in the decision-making behavior of the stakeholders. Contrastingly, it can be equally valuable to understand and model deviations from expected behaviors and other psychological specialties. Doing so can open up new avenues for improvement in product attributes and lead to improved customer satisfaction. It has been known that mathematical models with underlying assumptions different from what we encounter in real life can still be extremely valuable for both *normative and descriptive purposes.* One such example is preferential and utility independence of the multiple considered attributes [8]. This allows us to aggregate the technical and economic characteristics of alternatives into a single utility function in a relatively straightforward way. While these independence assumptions can be violated, the model is invaluable in its widespread applications [9]. Alternative methods of engineering decision making such as AHP, Pugh’s method, simple weight, and rate methods while inviting critique from some sources, still enjoy significant prevalence in engineering practice from their relative ease of use [10,11,12]. In engineering design and decision making under uncertainty, we often see the assumptions of normality or independence of random variables. It is commonly observed that these assumptions make problems tractable, allowing us to gain insights not possible otherwise.

More recently, there has also been much emphasis placed *on the cognitive aspects* involved in the design process and also those encountered during the use of products [13]. For example, Maier et al., [14] discuss cognitive-assistant (CA) facilitated design ideation groups where they compare and contrast human-assisted and CA-facilitated design. Brownell et al., [15] discuss the impact of individual team member proficiency in design. Ahmed and Demirel, [16] examine human-centered prototyping strategies in early stages of engineering design. Chou et al., [17] present a stakeholder agreement metric (SAM), which is based on the distance between the designs produced using different preference structures. Ahmed et al., [18] compare different prototyping methods in the early stages of product design when uncertainty effects are more pronounced. Autonomous products require users to relinquish decision-making control to some degree, and trust issues become a challenge in their adoption. Along these lines, Liao and MacDonald, [19] investigate the effect of manipulation of users’ trust in autonomous products. Slon et al. [20] investigate the challenging task of fast-paced decision making by engineers with conflicting preferences in manufacturing systems in the context of Industry 4.0. Changarana et al., [21] investigate designers’ ability to understand the thoughts, feelings, and psychological behavior of customers. Clearly, if human cognition affects and is affected by the design decisions, it merits investigation regardless of whether accepted normative processes are always followed.

In this paper, we cater to the existing literature on engineering design and customer experience by presenting experimental evidence on topics related to personal vehicle-related assessments. Our results show that these assessments are subject to inconsistencies and can exhibit destructive interference in decision-making probabilities. Subsequently, we propose a quantum-like (QL) model to describe the data, using the formalism of quantum mechanics. In our QL model, the uncertainty states and observables (questions) are represented as vectors and operators, respectively, in an abstract complex Hilbert space. We argue that in situations when multiple attributes or symptoms in the system are not independent, QL models can naturally accommodate the peculiarities such as the question-order effect (see, e.g., Pandey and Basieva, [22]). It is also demonstrated that complex-valued probability amplitudes admitting positive or negative interference are well-suited to reflect complicated dependence between the system attributes or symptoms that need to be reconciled together.

Our paper also contributes to the rapidly developing stream of research on applications of the formalism and methodology of quantum theory outside of physics, especially to decision making. This framework is known as quantum-like modeling to distinguish it from extended research on the coupling of cognition and consciousness with genuine quantum physical processes in the brain (see, e.g., [23,24,25]). In the quantum-like framework, humans are considered macroscopic quantum information processors. One of the sources of informational quantumness is contextuality of human perception, attention, and emotions. We recall that contextuality is one of the basic features of quantum theory, which is actively investigated in quantum information theory. Contextuality is closely coupled with another distinguishing feature of quantum observables—the presence of incompatible observables (or, generally, the Bohr’s complementarity principle). Such observables generate events, for which the logical operation of disjunction—if at all applicable—can lead to behavioral inconsistencies and paradoxes. By rapidly moving from one cognitive context to another, the decision maker prefers to transfer information in the state of superposition of a few alternatives. The resolution of superpositions and construction of a classical probability model can be time-consuming and have high demand on computational resources. Such processing of superpositions—updating corresponding quantum states and not probability distributions—is valuable from the viewpoint of information processing, especially in the situation of information overload generated by the complexity of the decision process and time constraints [26]. The latter is especially important in engineering, where projects have strict deadlines, and product alternatives must be evaluated over multiple competing objectives.

The framework of Contextuality-By-Default [27] can account for virtually any statistical data. It provides individual labels for variables in every context and then analyses the magnitude and consistency of the context’s influence. In contrast, we focus on a “combination” of contexts and their influences on the statistics. 

During the last few years, quantum-like approaches flourished by attracting researchers from various fields, cognition, psychology, decision making, economics, game theory, and (as in the present paper) engineering, as well as physics and mathematics (See, e.g., [28,29,30,31,32,33,34,35,36,37,38,39,40,41,42,43,44,45]) For example, quantum interference effects and conditional probabilities in decision making have been analyzed in [46,47] in the framework of quantum-like Bayesian Networks to model “irrational” decisions. We would like to note that it is conditional probabilities that we have in mind when we speak about the “causes” of some phenomena. We do not consider causality in time nor causal mathematical probabilistic frameworks as was proposed by Pearl [48]. 

As a final note for this section, it may be tempting to think that the events whose probabilities we are considering are necessarily subjective in nature. Classical probability has been used successfully in many design-projects under uncertainty settings, where problems are well understood and probabilities of events and their conjunctions can be calculated exactly [49,50]. However, for most engineering problems, subjective assessment of probability is still a major ingredient. This is particularly true for systems with many connections and interactions between its constituents. Two examples that best serve to explain this point are the Tenerife air disaster and the GM ignition switch issue [51]. For complex systems such as air transportation and automobiles, it is not always possible to assess joint probabilities of events and their impact on the system as a whole. In quantum information theory, philosophical considerations of limited access to probabilities stimulated the creation of the subjective probability interpretation of quantum mechanics, Quantum Bayesianism (QBism), see, e.g., Fuchs. Applications of QBism in decision making were discussed in [52].

This paper is organized as follows. In the following section, we present the background discussion and the motivation behind the proposed approach. Section 3 presents the mathematical formulation for the quantum-like model that we propose. This is followed in Section 4 by the design of the survey and analysis of the results. Section 5 concludes and provides avenues for future research.

## 2. Background and Motivation

Rational decision makers are expected to estimate probabilities and update their preferences according to Kolmogorov’s axioms and the Bayes rule of classical probability theory (Oaksford & Chater, 2009) [53]. A thorough analysis of cognitive processes and biases of rational agents on multiple levels can be found in [54,55]. Simply put, for the case where there is some value from a design at stake, monetary or otherwise, being “rational” can be defined as making decisions that maximize expectation value (or utility) from the outcome. Mathematically speaking, it has been advocated that a customer or a designer should seek to maximize a performance function, usually their utility function (or its expectation when uncertainty is present) while making decisions:(1)$$x^{*}=\underset{x}{argmax}\left(E\left[U\left(Y\left(x,q\right)\right)\right]\right)=\underset{x}{argmax}\left(\int_{D}^{}U\left(y\left(x,q\right)\right)f_{Q}\left(q\right)dq\right)$$
where x is the vector of decisions, Q is the vector of uncertain parameters, q is its realization, while D is the support of the distribution of Q**,** i.e., the set of values where the pdf is non-zero. Y(.) is the vector of attributes and U(.) is the utility function. It is often seen that decision makers deviate from the above tenet of formal decision making.

Naturally, people may have other values and interests apart from the attributes directly under consideration as described above. It is not always possible to frame a decision problem such that all externalities are made irrelevant. It is also possible that decision makers do not properly apply probability theory or simply make mistakes when calculating. Furthermore, there is also the case of limited resources, limited information, and bounded rationality, all of which affect decisions. The resulting seemingly “irrational” behavior gives rise to “paradoxes” that have been thoroughly investigated in the field of cognitive psychology and behavioral economics [56,57,58].

Quantum-Like (QL) approaches have also been widely used in describing these paradoxes [29,30,59,60,61]. The approach is used to calculate probabilities and describe statistics collected in the experiments are described by the mathematical formalism of quantum mechanics. It is important to note that QL approaches are an operational formalism and should be distinguished from the idea that “free will” and choices of a person have their foundation in the actual quantum physical processes in the brain.

Quantum-like formalism is naturally suited to model deviations from classically predicted behavior because, basically, it is richer than classical probability theory. For example, it admits an interference term in the law of total probability (LTP) when formulated using quantum formalism. In Equation (2), we consider LTP for a probability of a decision (A=a) conditioned upon two possible disjoint events/causes $\left(b_{0},b_{1}\right)$:(2)$$p\left(A=a\right)=p\left(A=a|B=b_{0}\right)p\left(B=b_{0}\right)+p\left(A=a|B=b_{1}\right)p\left(B=b_{1}\right)\;+Interference\;Term$$

In quantum formalism, probabilities are calculated as squared norms of ψ-function projections, as given by the Born rule, e.g., [62]. Probability amplitudes have associated phases that can lead to both negative and positive interference between terms. It has been shown that non-commuting observables account for the question order effect which explains how respondents give different answers based on the order of questions asked [39]. In Haven and Khrennikova, [63] QL models were used to describe a violation of LTP in financial decision making by investors. It was also shown that the projection postulate of QM easily accommodates an update from zero prior [64]. The main convenience is that QL models use explicit rules for the state update, not only for the probability update. In classical probability theory, Kolmogorov axioms and Bayesian update rule are used. In principle, if we postulate a rule for updating our (classical) state following disturbance provided by the measurement (say, by giving an answer to a previous question), we would be able to produce absolutely the same results as quantum formalism [65]. Alternatively, any single experiment can be thought of as taking place in its own unique context and possessing its own unique set of random variables, the so-called Contextuality-by-Default notion [66]. No classical joint probability distribution can be assigned to the variables from mutually incompatible contexts. The literature on CbD usually considers joint measurement of a few pairs of questions, one question playing the role of context for the other. A cycle of pairwise joint measurements is needed to pinpoint the presence of incompatibility. In contrast, in the present work, we consider not joint but sequential measurements and study the effect of combination of a few conditioning questions. Let us revisit the possible violation of the law of total probability as described in Equation (2). It is possible to design an experiment where an unconditional choice is not a weighted sum of the few conditional choices. The quantitative measure of this violation is the interference term. Furthermore, it has been shown in recent work that the interference term even in quantum formalism has its own restrictions. This phenomenon becomes more interesting with an increasing dimensionality of the problem state space [67]. 

In its strongest form, this effect presents itself as a violation of Savage’s *Sure Thing Principle* [68]. In Equation (2), even if we admit that the probability distribution p(B) is different in the LHS and the RHS, the LTP is still found to be violated. That is, judgement about A in the state of uncertainty about B (LHS) is very different from both cases on the *RHS,* where either $B=b_{0}$ or $B=b_{1}$ are guaranteed. A well-known example of the violation of the “sure thing principle” is the prisoner’s dilemma, where each of the two prisoners is motivated to betray the other if they know the other’s action. In either case, the rational choice is to betray. However, it has been seen that when a prisoner does not know the other’s decision, they do not betray. An interesting treatment of this problem was given in [34]. The authors consider dynamics of the mental state via classical Kolmogorov equations, as well as quantum evolution under some Hamiltonian. They demonstrate that the former cannot lead to the observed probabilities. A unitary evolution of quantum pure states is eternal, not quenching oscillations between “yes” and “no” as predicted from the work. A more sophisticated evolution under the Lindblad equation can account for stabilization of opinion [69,70,71]. Apparently, it is also possible to construct a more sophisticated classical equation, with more parameters and taking into account other motives of the participants. In this paper, we do not consider the dynamics of mental states and limit our considerations to the simplest case of projections and pure initial states with the goal of illustrating the effect of complex probability amplitudes amplifying or quenching each other.

The objective of this work is to highlight some of the cognitive inconsistencies that can occur in engineering decision making and apply a QL formalism for their description. For this purpose, we perform a survey where we ask the participants simple engineering questions. It is shown in the results that quantum formalism can explain the responses we received, while straightforward classical probability calculations are not suitable. Consider two events. In a classical measure-theoretical model, if each of the events independently increases the probability of some third event, then their combination should increase this third event’s probability even more in line with Bayesian probability update. This is not necessarily the case in quantum formalism, and our experiment is designed to illustrate this point.

A related example comes from quantum computing, for example, in Shor algorithm. An entangled state obtained after the application of a function to the superposed (but factorizable) initial state can be considered a combination of disparate ways to achieve the same outcome. These ways may be combined in the way of both positive and negative interference. In decision making, two motives or incentives for the same decisions, or two pieces of information, each one favoring the same outcome, may work the opposite if considered together. This fits the negative interference feature, provided by quantum formalism but not by classical probability. 

The goal of the survey experiments we conducted was not merely to pinpoint possible “fallacies” in engineering decision making. They are unavoidable and have been ignored so far, seemingly without much consequence. So, our aim is not to try to show the inapplicability of classical probabilistic models. We argue that another simple and linear mathematical model based on quantum formalism can be a prospective tool for analysis and even, hopefully, prediction of these ubiquitous “paradoxical” effects. Hopefully, someday we will be able to say in advance that certain experimental settings are prone to certain type and magnitude of “inconsistencies”. Research in this domain may contribute to the refinement of the parameters of the quantum probability framework which at the same time need to be sufficiently robust to be applicable in a significantly wide range of experiments. This may lead to better techniques for capturing the preferences of decision makers and predicting their actions under different operating scenarios. As a result, better products, systems, and services could be designed by engineering firms.

## 3. Preliminaries

Consider an engineer attempting to troubleshoot a possible problem with a vehicle given some limited information. Throughout the paper, we consider only binary variables. Furthermore, consider two possible **independent** causes or symptoms, *a* and *b,* to some end effect that is observed, such as the malfunction indicator light (MIL), also referred to as the “check-engine light”, on the dashboard. We denote this end-effect by *d*. We can write:(3)p(d|a)p(a)=p(d∩a)=p(a|d)p(d)

Similarly:(4)p(d|b)p(b)=p(d∩b)=p(b|d)p(d)

From the independence of the two, we have:(5)p(a∩b)=p(a)p(b)

Importantly, for our discussion we assume that *a* and *b*
**conditioned** on *d* are also independent. The survey does not collect data to test this assumption. We can reasonably assume this in certain cases, e.g., low gas and low oil are independent conditions, including the case when there is an unknown light indicator on the dashboard. Therefore:(6)p(a∩b|d)=p(a|d)p(b|d)
(7)p(d|a∩b)p(a∩b)=p(d|a∩b)p(a)p(b)=p(a∩b|d)p(d)=p(a|d)p(b|d)p(d)

Individually, *a* or *b* increasing the probability of *d* implies: (8)p(d|a)>p(d), p(d|b)>p(d)

Hence:(9)p(a|d)>p(a), p(b|d)>p(b)

Substituting in (7), we obtain p(d)<p(d|a∩b), which implies p(a∩b|d)>p(a∩b).

We have shown that for conditionally independent a and b, if both a and b increase the probability of d, then their intersection a∩b also increases the probability of d. This is true for classical probability space but not for quantum. Below, we will provide a quantum model which exhibits different behavior. In real life, it is easy to imagine situations when each of the causes increases the probability of some event, but taken together, two causes cancel each other out and may even result in a probability lower than the original one. For example, if we consider potential causes for the check-engine light, one cause is enough to provide confidence about the meaning of the indicator, but two or more causes can have “negative interference”, which may be so large that it would leave the engineer even more confused than they were originally.

Let us now briefly consider the following question: How safe is our assumption of conditional independence? We could look out for alarming signs.

Let us denote negation of *d* by $\bar{d}$.

**Lemma** **1.**
*The inequality*

(10a)
p(d|a∩b)<p(d)

*is equivalent to the inequality*

(10b)
$$p\left(a\cap b|d\right)<p\left(a\cap b|\bar{d}\right).$$



**Proof.** Set x=p(a∩b∩d) and y=p(d). Then, (10b) can be written as: x/y<(p(a∩b)−x)/(1−y), i.e., x<p(a∩b) y or x/(a∩b)<y. The latter inequality coincides with (10a). This works the other way, too, from (10b) to (10a). □

If inequality (10b) holds, then one can easily construct a classical probability space accounting for the situation:(10c)p(d|a)>p(d), p(d|b)>p(d), p(d|a∩b)<p(d)

On the other hand, violation of (10b), i.e., validity of inequality:(10d)$$p\left(a\cap b|d\right)>p\left(a\cap b|\bar{d}\right)$$
definitely makes the situation (10c) impossible.

The crucial point is that the Lemma holds true only within the classical probability model. In the quantum probability model, conditions (10a) and (10b) are not equivalent. Even under condition (10d), opposite to (10b), we can obtain situation (10c). 

Let us now consider a quantum-like model. Let us now assign to the symptoms the projectors A, B, D, such that AB=BA is also a projector, [A,B]=0, [AB,D]≠0. Then, for a pure initial quantum state ψ, we have:$$p_{\psi }\left(AB|D\right)=||ABD{\psi ||}^{2}/||D{\psi ||}^{2}$$
$$p_{\psi }\left(AB|\bar{D}\right)=||AB\bar{D}{\psi ||}^{2}/||\left(I-D\right){\psi ||}^{2}$$
$$p_{\psi }\left(D|AB\right)=||DAB{\psi ||}^{2}/||AB{\psi ||}^{2}$$
where the index ψ denotes the probability with regard to the pure quantum state ψ. 

The LTP (2) now takes the form: $||AB{\psi ||}^{2}=||ABD{\psi ||}^{2}+||AB\bar{D}{\psi ||}^{2}+\Delta$, where the interference term is $\Delta =||AB{\psi ||}^{2}-x-||AB\bar{D}{\psi ||}^{2}$, and we introduce as before $x=||ABD{\psi ||}^{2}$ and $y=||D{\psi ||}^{2}$. The interference term is closely connected to the question order effect. If we define order effect as:$$OE\left(D,AB\right)=p\left(D|AB\right)p\left(AB\right)-p\left(AB|D\right)p\left(D\right)=||DAB{\psi ||}^{2}-||ABD{\psi ||}^{2}$$
we can see, e.g., Pandey and Basieva [22] that $\Delta =OE\left(D,AB\right)+OE\left(\bar{D},AB\right)$.

Now, $p_{\psi }\left(AB|\bar{D}\right)=\frac{||AB{\psi ||}^{2}-x-\Delta }{\left(1-y\right)}$ and inequality (10b) can be written as:$$\frac{x}{y}<\frac{||AB{\psi ||}^{2}-x-\Delta }{1-y},\;\;\mathrm{or}\;x\left(1-y\right)<\left(||AB{\psi ||}^{2}-x-\Delta \right)y,\;\;\mathrm{or}\;\frac{x}{||AB{\psi ||}^{2}-\Delta }<y.$$

We want to compare this inequality with (10a), which can be written as:$$\frac{||DAB{\psi ||}^{2}}{||AB{\psi ||}^{2}}<y$$

We see that both numerators and denominators of (10b) and (10a) are different unless AB and D commute. Hence, generally inequality (10b) does not imply (10a).

The corresponding counterexample from quantum probability calculus is presented in Figure 1, where magenta line p(D|AB) lays higher than the unconditioned black p(D), while the dashed black p(AB|D) is higher than dotted blue p(AB|NotD). Figure 1 demonstrates that even if we pay attention to avoiding the warning sign $p\left(a\cap b|d\right)<p(a\cap b|\bar{d})$, we can still account for the “paradoxical” situation (10c).

Generally, we think that it is reasonable to presume that minor faults in an automobile, such as low tire pressure or absence of windshield washer liquid, happen independently, and the indicator is responsible for one fault only.

It is generally difficult to check for these independence assumptions by means of surveys, and we did not include corresponding questions in our survey. Our point is that for such reasonable assumptions and plausible real-life situations, the straightforward application of classical probability theory is not satisfactory, while the QL approach can be more suitable. A difficulty that arises in problems with quantum conditional variables is that they are doubly stochastic, see, for example, [70]. This restriction comes down to p(a|b)=p(b|a). In this paper, we do not address this restriction, and our purpose is mostly demonstrational. This restriction is only for observables represented by Hermitian operators with non-degenerate spectra. If spectra are degenerate, the outcomes correspond not to eigenvectors but to eigen-subspaces; then, the quantum matrix of transition probabilities need not be doubly stochastic.

In our quantum-like model, we will consider two (and later three) commuting “conditioning” projectors A and B, denoting the independent events or symptoms, and the main projector D, which corresponds to the question “if it is safe to drive” and must not commute with A and B. A is acting in a two-dimensional subspace and B in a 3-dimensional Hilbert space:(11)$$A_{2}=\left(\begin{array}{cc} 1 & 0\\ 0 & 0 \end{array}\right),\;B_{3}=\left(\begin{array}{ccc} 1 & 0 & 0\\ 0 & 1 & 0\\ 0 & 0 & 0 \end{array}\right),\;A=A_{2}⊗I_{3},B=I_{2}⊗B_{3}$$
where *I* denotes identity matrices, and the subscripts indicate their dimensions. The projectors *A* and *B* commute, and their product is also a projector:(12)$$AB=BA=A_{2}⊗B_{3}$$

In the six-dimensional space, the initial “mental” state of our typical participant is described by a |ψ⟩, which can also be factorizable:(13)$$\left|\psi \right\rangle =\left(\begin{array}{c} \psi_{1}\\ \psi_{2} \end{array}\right)⊗\left(\begin{array}{c} \psi_{3}\\ \psi_{4}\\ \psi_{5} \end{array}\right)$$

In this case, our measurement operators *A* and *B* acting on such initial state are equivalent to independent observables. However, if we just want the condition p(a∩b)=p(a)p(b) to be true, it can hold for non-factorizable |ψ⟩ as well:(14)$$p\left(a\cap b\right)=||AB{\psi ||}^{2}=||BA{\psi ||}^{2}=||A{\psi ||}^{2}||B{\psi ||}^{2}=p\left(a\right)p\left(b\right)$$

It should be noted that this kind of independence places restriction only on absolute values of the first three elements of |ψ⟩. If each of the six elements of (not necessarily factorizable) |ψ⟩ is represented as $\psi_{j}=\sqrt{a_{j}}\;e^{i\phi }$, then $a_{1}=a_{2},\;a_{1}^{2}+a_{1}\left(a_{3}+a_{4}+a_{5}-1\right)+a_{3}\;\left(a_{4}+a_{5}\right)=0$ is required for the independence of *A* and *B*. Naturally, all $a_{j}$ must be positive real numbers by definition, which is another restriction.

We parameterize our main question, *D* “if it is safe to drive”, and the initial state |ψ⟩ as:(15)$$D_{6}=\left(\begin{array}{cccccc} 1 & i & 0 & 0 & 0 & 0\\ -i & 1 & 0 & 0 & 0 & 0\\ 0 & 0 & 1 & -i & 0 & 0\\ 0 & 0 & i & 1 & 0 & 0\\ 0 & 0 & 0 & 0 & 2\cos^{2}\left(\alpha_{1}\right) & 2\cos\left(\alpha_{1}\right)\sin\left(\alpha_{1}\right)\\ 0 & 0 & 0 & 0 & 2\cos\left(\alpha_{1}\right)\sin\left(\alpha_{1}\right) & 2\sin^{2}\left(\alpha_{1}\right) \end{array}\right)/2$$
(16)$$\left|\psi \right\rangle =\left(\begin{array}{c} \sqrt{a_{1}r}\\ e^{i\pi \theta }\sqrt{a_{1}\left(1-r\right)}\\ \sqrt{a_{3}}\\ -i\sqrt{a_{3}}\\ -\sqrt{a_{5}}\\ \sqrt{1-a_{1}-2a_{3}-a_{5}} \end{array}\right)$$

Then, obviously, $||DAB{\psi ||}^{2}=p(d|a\cap b)$. Naturally, this parametrization is not unique, and we tried to find the simplest yet non-trivial representation. At $a_{3}=a_{4}=0.15,\;a_{5}=0.1$*,* $\alpha_{1}=1.25\pi ,\;\theta =0.427\pi ,\;a_{1}^{2}+a_{1}\;\left(2a_{3}+a_{5}-1\right)+a_{3}\;(a_{3}+a_{5})=0,$ we can demonstrate that, individually, *A* and *B* can increase the probability of *D* even from zero, violating Cromwell’s rule, see [64]. At the same time, the *AB* projector decreases the probability of *D*, possibly back to zero. An important part of this demonstration is the values of *AB* conditioned on *D*—they are higher when conditioned on *D* than on the negation of *D.* This is a crucial difference from the classical description. 

In Equation (16), we use parameters $a_{1}$ and r rather than $a_{1}$ and $a_{2}$ for convenience. Figure 1 describes the phenomenon visually as the parameter r is varied (we focus on the region where r=0.5). Conditions *A* and *B* individually increase the probability of *D* even from values close to zero. However, we see that combined conditions *A* and *B* (magenta) have much smaller or no effect. Meanwhile, the probability of the combination *A* and *B* when *D* is true is significantly higher than the combination of *A* and *B* when *D* is not true. The classical explanation for the phenomenon—that our *A* and *B* conditioned on *D* have their intersection primarily in the domain of not *D*—does not hold. We do not presently model the exact independence of *A* and *B* conditioned on *D.* However, even avoiding this kind of dependence, we can have $p\left(a\cap b|d\right)>p(a\cap b|\bar{d})$ and still reproduce the phenomenon p(d|a)>p(d), p(d|b)>p(d), p(d|a∩b)<p(d).

## 4. Results and Discussions

The survey we designed consisted of randomized questions aimed at individuals with some engineering background, even though it was not required. Different groups of respondents received different sets of questions through randomization to monitor the effect of priming. The questions pertained to some background questions in engineering, followed by questions regarding preferences for different types of vehicles (electric, gasoline, autonomous) followed by a probability assessment question. In this paper, we address only the probability assessment question(s) listed in Table 1, as they are relatively independent of the other questions asked to the respondents. In this question, one-half of the respondents are asked to provide the probability that a vehicle with an unknown check-engine light indicator on is drivable (D) independently and then given a series of symptoms, A,B, and C. The other half of the respondents were given all the symptoms together and were asked if the vehicle is drivable. We are not concerned with drivability when there is no light indicator on, so we use a short notation *D* rather than *D|L* for drivability conditioned on presence of the light (*L*) indicator.

As of the writing of this paper, 76 total responses were observed which are summarized (averaged probability numbers provided by the respondents) in Table 2. The number of responses were almost evenly split between the two groups.

In our experiment, we deal with three conditioning pieces of information as opposed to the two presented in the discussion in Section 3. For the modeling part, we introduce a new observable *C,* such that $C_{2}=A_{2}$*,* and use the same *A, B,* and *D* raised to a higher dimension to accommodate the specific problem’s characteristics. We also use for $\left|\psi \right\rangle =|\psi_{6}⊗|\psi_{2}\rangle$, where $\left|\psi_{2}\right\rangle$ is the new part of the state in the subspace where *C* acts, and $\left|\psi_{6}\right\rangle$ is similar to the initial state in previous considerations, with θ=1.5π:(17)$$\left|\psi_{6}\right\rangle =\left(\begin{array}{c} \sqrt{a_{1}r}\\ -i\sqrt{a_{1}\left(1-r\right)}\\ \sqrt{a_{3}}\\ -i\sqrt{a_{3}}\\ -\sqrt{a_{5}}\\ \sqrt{1-a_{1}-2a_{3}-a_{5}} \end{array}\right),\;\left|\psi_{2}\right\rangle =\left(\begin{array}{c} \sqrt{r_{2}}\\ e^{i\pi \theta_{2}}\sqrt{1-r_{2}} \end{array}\right)$$

Similarly, the operators are: $A=A_{2}⊗I_{3}⊗I_{2},\;B=I_{2}⊗B_{3}⊗I_{2},\;C=I_{2}⊗I_{3}⊗C_{2},$ and *D* is the direct product of previous operator and the new, two-dimensional one: $D=D_{6}⊗D_{2}$, where $D_{6}$ is defined by (15) and
(18)$$D_{2}=\left(\begin{array}{cc} \cos^{2}\left(\alpha_{2}\right) & \cos\left(\alpha_{2}\right)\sin\left(\alpha_{2}\right)\\ \cos\left(\alpha_{2}\right)\sin\left(\alpha_{2}\right) & \sin^{2}\left(\alpha_{2}\right) \end{array}\right)$$

This factorizable form of the operators and the |ψ⟩’s ensure that they commute, and their product is also an operator. This ensures that the joint event of, *A, B,* and *C* is defined.

We modified the parameters to fit the observed data, and the following values provide a good fit:$$r=0.5,\;\alpha_{1}=0.75\pi ,r_{2}=0.01,\;\alpha_{2}=1.268\pi ,\;\theta_{2}=0.48\pi ,\;a_{3}=0.15,\;a_{5}=0.08.$$

Our conjecture for this experiment was that there are situations where conditioning on similar kind of information which drive priors in one way, when combined, may drive the prior in the opposite way. The results from the fitted data and Figure 2 shows that this is indeed the case. We see from the data and the figure that conditioning on single piece of information increases the confidence of safe driving from prior *p (D)* to higher values given by *p (D|A), p (D|B),* and *p (D|C)* (Again, we focus on the region where r=0.5). However, when all three (independent) pieces are combined, it leads the prior in the opposite direction, that is, p(D|ABC)<p(D) at *r =* 0.5. While this result may be explained using reasoning that requires additional assumptions or knowledge, it cannot be modeled classically using just the information given, and quantum formalism provides a straightforward solution.

## 5. Conclusions and Future Work

Many studies in psychology, business, and engineering have shown that human decision makers routinely deviate from the tenets of classical probability. This is true while making decisions and also when assessing probabilities of events, and it is especially true for cases where probabilities of intersections and unions of events must be assessed. Since such problems arise naturally many times in decision making, deviations are also encountered. Clearly, better modeling of such deviations has the potential to impact the quality of decisions made, as well as the ability of products and systems to be designed such that they maximize customers’ satisfaction. While it may be tempting to expect or even demand consistency with classical probability in all the assessments, it is equally beneficial to consider altogether different models of probability such as quantum probability. The quantum-like (QL) models that utilize formalisms from quantum mechanics have been used extensively in the psychology literature to explain deviations from classical probability predictions. In engineering design, the impact of using proper models can be the difference between products and systems that best satisfy the requirements and ones that fail. 

In this work, we presented results from a survey conducted regarding the effect of multiple observed symptoms on the drivability of a vehicle. Problems such as these are routinely encountered in real life and while making engineering decisions. The complexity of today’s systems necessitates that many such assessments are made using judgement and prior experience as opposed to rigorously. We demonstrate that the set of responses that we collected cannot be explained using classical probability. The quantum formulation presented easily models it, as it allows for both positive and negative “interference” between events. In particular, it was seen that different symptoms individually increased the probability of a vehicle’s “drivability” event from the prior. However, when the symptoms were put together, the prior remained more or less unchanged. This result cannot be modeled classically in a direct way unless additional information is obtained or other assumptions are made. 

In future work, we would investigate other related effects as well, such as order effects and various paradoxes noted in the literature, particularly from the context of engineering decision making and engineering design. We would also investigate if the parameters derived for the quantum model in one decision situation can be translated to another. 

## Figures and Tables

**Figure 1 entropy-24-00725-f001:**
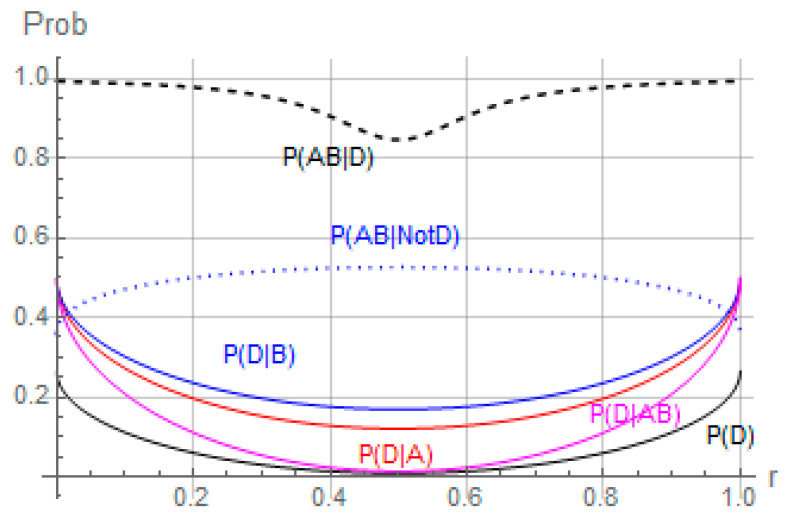
Conditions *A* and *B* (red and blue) increase the probability of *D* (black), even from values close to zero. Combined conditions *A* and *B* (magenta) have smaller or no effect. Meanwhile, probability of the combination *A* and *B* when *D* is true (black dashed line) is significantly higher than the combination of *A* and *B* when *D* is not true (blue dotted line).

**Figure 2 entropy-24-00725-f002:**
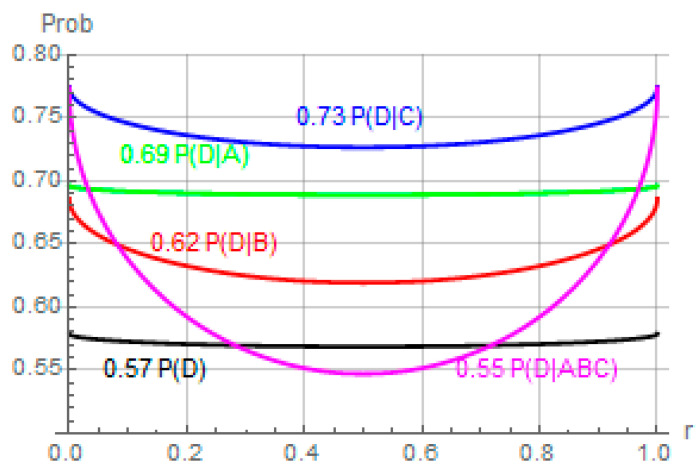
Prior and conditional probabilities fit to the experimental data.

**Table 1 entropy-24-00725-t001:** Subset of Questions Asked in the Survey Used in the Analysis in This Paper.

Respondent Group 1	Respondent Group 2
**Question on uninformed prior about safety to drive (D):** 1D. You are driving a certain car for the first time and notice that an unknown light is lit on the dashboard. How confident are you that the car can be driven safely in this situation? Please provide the probability as a percentage.	**Posterior decision based on all three conditions ABC:** 2D. You are driving a certain car for the first time and notice that an unknown light is lit on the dashboard. You know there is no windscreen washing liquid, tire pressure is slightly below the prescribed value, and 4-wheel drive is on from the way the car is handling. How confident are you that the car can be driven safely in this situation? Please provide the probability as a percentage.
**Decisions with three different conditionals A, B, or C:** 1A. You are driving a certain car for the first time and notice that an unknown light is lit on the dashboard. You know there is no windscreen washer liquid. How confident are you that the car can be driven safely in this situation? Please provide the probability as a percentage. 1B. You are driving a certain car for the first time and notice that an unknown light is lit on the dashboard. You know that the tire pressure is slightly below the prescribed value. How confident are you that the car can be driven safely in this situation? Please provide the probability as a percentage. 1C. You are driving a certain car for the first time and notice that an unknown light is lit on the dashboard. You know that 4-wheel drive is on from the way the car is handling. How confident are you that the car can be driven safely in this situation? Please provide the probability as a percentage.	

**Table 2 entropy-24-00725-t002:** Summary of the Responses Received for the Questions in Table 1.

Group 1 Responses		Group 2 Response	
1D.	0.57	2D.	0.55
1A.	0.69		
1B.	0.63		
1C.	0.73		

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
