# Peer review of "More Causes Less Effect: Destructive Interference in Decision Making"

_entropy, 2022, doi:10.3390/e24050725_

Round 1

Reviewer 1 Report

I find this paper interesting and recomment publishing it.

Nevertheless, I recommend adding to the list of references the paper Rashkovskiy, S., & Khrennikov, A. (2020). Psychological ‘double-slit experiment’in decision making: Quantum versus classical. Biosystems195, 104171, which deals with similar issues.

Author Response

Thank you very much for the review. We have added the reference to the paper as you suggested. We also modified the formulas for quantum conditional probabilities for a pure state.

Reviewer 2 Report

The article’s main contribution is extending the Q-Like (QL) approach, which have proven to be effective elsewhere, to the engineering practices and dealing with such products as airplanes or automobiles, and in arguing why this approach is likely to be more effective than classical-like probabilistic approaches in this area as well. It could be published nearly as is. I do have a few suggestions that the authors might want to consider.

I think that the notions of contextuality and (which is a special concept) “contextuality by default” have a specific (mathematized) meaning in quantum theory and QL theories, and should be explained, if briefly, a bit further, perhaps by giving an example. It is not only that “any single experiment can be thought as taking place in its unique context, and possessing its own set unique set of random variables” (p. 5), which is true (the phrase does not need a comma after “context”), but also that there are mutually exclusive contexts. This mutual exclusivity (also related to complementarity in quantum physics) precludes defining jointly certain random variables simultaneously in the way it is possible in classical theories. This aspect of contextuality is relevant to the article’s argument, which deals with a possible combination of causes and thus potentially contexts. Considering this subject might, however, extend the article beyond its scope. So, I do not insist that the authors address it.

p. 4, It is important to note that QL approaches are an operational formalism ...

I would replace this with “it is important to keep in mind,” because this point was already mentioned earlier.

p. 4, I don’t think that the reference to Sakurai’s book is necessary, as Born’s rule is standard. Or: (e.g., Sakurai 1985).

p. 6, We argue that another, simple and linear mathematical model, based on quantum formalism can be a prospective tool for analysis and even, hopefully, prediction of these ubiquitous “paradoxical” effects.

This is a bit unclear. Do the authors mean “practical predictions” here, given that QL-models are essentially probabilistically predictive, and hence their analysis already concerns predictions?

p. 7, For example, if we consider potential causes for the check-engine light, one cause is enough to provide confidence about the meaning of the indicator, but two and more causes can have “negative interference” and leave the engineer even more confused than they were originally.

Would this “negative interference” (which is indeed possible and can make one more confused than a single cause) necessarily make the engineer more confused than the absence of any known cause, which is often the original situation?

There are a few minor glitches in phrasing here and there, but they are minor. As I said, in general it is a lucidly written article.

Author Response

Thank you for the detailed review and very valuable suggestions.  

  • We have added the following details on CbD: “No classical joint probability distribution can be assigned to the variables from mutually incompatible contexts. The literature on CbD usually considers joint measurement of a few pairs of questions, one question playing the role of context for the other. A cycle of pairwise joint measurements is needed to pinpoint the incompatibility. In contrast, in the present work we consider not joint but sequential measurements and study the effect of combination of a few conditioning questions.”
  • We have modified the wording of our statement and reference on page 4 according to your suggestions.
  • Yes, you are right, negative interference does not necessarily make the situation worse than it was before any conditioning. We corrected the phrase to “… two and more causes can have “negative interference”, which may be so large that it would leave the engineer even more confused than they were originally.”
  • When we speak about prediction about “paradoxical” effects, we mean not a probabilistic prediction for this particular experiment, but rather extension to different – but may be similar – Ideally, we could be able to predict the value of interference term with some accuracy. We added the following sentence to clarify this point: “Hopefully, someday we will be able to say in advance that certain experimental settings are prone to certain type and magnitude of “inconsistencies”. Research in this domain may contribute to the refinement of the parameters of the quantum probability framework which at the same time need to be sufficiently robust to be applicable in a significantly wide range of experiments.”

Reviewer 3 Report

The authors attempt to compare the relation between classical and quantum conditional probabilities of three events. This is an interesting question from the point of view of applicability of quantum techniques to human decision making. Unfortunately, from the very beginning the manuscript contains principal deficiencies. 

Thus in line 307, where the authors deal with quantum probabilities, we read  "Let A, B, D be orthogonal projectors". It is not defined in what sense they are orthogonal. If these projectors are mutually orthogonal, then AB = AD = BD = 0 and the following formulas degenerate to trivial zeroes.  If they are not mutually orthogonal, then to what objects are they? 

In line 309, the authors claim to write a conditional quantum probability
p(AB|D) = Tr(ABD \rho)/ Tr(D \rho). However, the conditional quantum 
probability has the von Neumann-Luders form p(AB|D) = Tr(DABD\rho)/Tr(D\rho), which is basically different from the used expression. Therefore all following consideration, based on the incorrect formula, is not acceptable.    

In line 312, there is a mistake in the last equation, where instead of Tr(AB), there should be Tr(AB\rho). 

Concluding, although the considered problem could be of interest, but, 
unfortunately, the consideration is based on erroneous equations. All formulas throughout the manuscript have to be reconsidered using the correct von Neumann-Luders probability and cleaning out mistakes.   

Author Response

Thank you very much for the review and the crucial observation of the mistake. We have modified the formulas, but the conclusion stays the same – that the straightforward equivalence of the classical probability expressions does not hold in quantum formalism. Now we limit our consideration to pure states, so the formulas in this part are now corrected and, actually, coincide with those used in the remainder of the work, e.g., in the modelling section.

Regarding orthogonality of the projectors: no, it is not mutual orthogonality, we deleted the word “orthogonal” to avoid the confusion. The A, B, D projections correspond to the same symptoms (conditions) as a, b, and d above, in the classical scheme.

Please see the corrected manuscript or the excerpt in the attachment.

Round 2

Reviewer 3 Report

The error in the von Neumann-Luders probability, made in the first variant 
of the manuscript, has been corrected. Unfortunately, a new mistake is made.

In the last line of page 7, the expression for p(AB|bar{D}) contains an error.
In the right-hand side of this expression the term (-x) has to be replaced by x. This gives different formulas in page 8, which need to be corrected. Otherwise, the expressions in the lines 1,2, and 4 from the top of page 8 are mistaken. 

Author Response

Dear Reviewer, thank you very much for the attention. I added some clarification to this problematic part of the manuscript. I am sorry if I missed your point. At any rate I am happy that the formulas for conditional probabilities are correct. 
